

**Study on the effects of storm movement on rainfall-runoff modelling at the basin scale**
Shahram KhalighiSigaroodi[1,3], Qiuwen Chen[2,3,*]
1. Faculty of Natural Resources, University of Tehran, Iran
2. CEER Nanjing Hydraulics Research Institute, Nanjing, 210023, China
3. RCEES Chinese Academy of Sciences, Beijing, 100085, China
* Correspondence to: Tel./Fax: +86 10 62849326, E-mail: qwchen@nhri.cn



**Abstract:** A number of studies have emphasized the effects of rainfall movement on runoff

simulation; nevertheless, due to the lack of rain gauges inside sub-basins, a method using a

hyetograph of the nearest gauges to a sub-basin is usually employed. This study investigated

the negative effects of neglecting rainfall movement on overland simulation results in even a

middle-sized basin. Simulations were carried out under two conditions: (1) stationary

conditions where the nearest gauge hyetograph was used and rainfall movement was ignored,

which is quite common in case of a lack of data; (2) moving conditions where a shifted

hyetograph based on hyetograph timing recorded in the basin was used. The simulation

results were compared with the measured discharge at the outlets. The results revealed that

using the shifted hyetograph, which could consider the rainfall movement over sub-basins,

decreased the mismatches between the simulated and observed hydrograph. In some of the

cases, the shifted hyetograph reduced the relative difference more than 20%.



## 1. Introduction

Since the first reports in the 1960s (Maksimov, 1964: Yen and Chow 1969) emphasized that

higher peak flows are generated whenever the precipitation moves from upstream toward

downstream, and conversely, rainfalls passing from down to upstream result in a rounded

hydrograph, a great deal of research has investigated the effects of rainfall movement on the

shape of the runoff hydrograph in the past half century. Most studies (Ngirane et al., 1985;

Singh, 1997, 1998) have applied mathematical approaches to obtain a better understanding of

the effects of storm speed and direction characteristics on the hydrograph shape. Their results

showed that hyetograph characteristics, such as rainfall pattern, duration, intensity, direction

and speed, significantly affected the hydrograph shape. Some researchers (Singh, 1998;

Mizumura, 2011) adopted a kinematic wave equation to model the hydrograph in the case of

a moving rainstorm. Their results showed that the maximum flow depth was generated when

the rainstorm speed equalled the flood movement toward the outlet, and the speed of the

storm had a greater impact for larger Manning's roughness coefficients. Recent studies have

preferred dynamic wave models based on Saint Venant equations to obtain flexible results

under varying conditions (Costabile, 2012). Kim and Seo (2013) applied a dynamic wave

model base on shallow water equations to study the effects of storm movement on runoff

generation in a V-shaped watershed experimentation system. The results revealed that storm

movement could generate a loop in the stage-discharge curve, and changes in storm

movement direction could invert the rotation of the loop. In addition, there has been some

research (De Lima et al. 2002) using rainfall simulators at laboratory scale to investigate the

effects of storm movement. Laboratory portable rainfall simulators and flumes were used to

simulate the hydrograph response to moving storms and subsequently soil erosion (De Lima

et al. 2003). They applied different hyetograph patterns to study the effects of rainfall

characteristics on the runoff hydrograph. The simulation outputs of hypothetical storms



moving upward and downward over a laboratory impervious plane revealed that the peak
discharges and hydrograph shape were highly affected by storm movement. Saghafian et al.
(1995) used a two-dimensional runoff model and a Monte Carlo method to investigate storm
movement effects on runoff. The results indicated that when storm movement is slow, a
stationary rainstorm could be used in simulations; while when storm movement is fast, a
stationary rainstorm was not acceptable. Ogden et al. (1995) showed that the runoff
hydrograph was more sensitive to storm speed than direction in two-dimensional basin
topography. Base on Manning's equation, the peak maximum occurred when the storm
moved toward downstream at a critical speed equalling half the flow velocity.
Although there is well-known background on the effects of moving storms on overland
flow generation, most of the interest has focused on laboratory experiments (Singh, 1997,
1998; De Lima et al. 2002, 2003) or mathematical approaches (Costabile, 2012; Kim and Seo
2013; Saghafian et al., 1995, Ogden et al., 1995). These studies emphasized the effects of
movement on runoff generation via a synthetic hyetograph whose direction, speed and
intensity were well-controlled by the researchers. However, few studies are available about
rainstorm movement effects on runoff in natural environments of real basins, especially in the
case of data deficiency. The objective of this study was to (1) precisely examine the effects of
moving storms on hydrograph simulation at the basin scale using natural recorded rainfall-
runoff; (2) provide an approach to determine the rainfall characteristics under the conditions
of data shortage in ungauged basins.
**2. Materials and methods**
**2.1 Study area and data availability**
Barandoozchay basin, one of the Urmia Lake sub-catchments, is located in the northwest of
Iran. The study area lies in between Urmia Lake and the Iran-Iraq-Turkey international





border from 44° 45' E to 45° 14' E and 37° 06' N to 37° 29' N. The area of the basin is about
1146 km$^2$.
The basin is divided into 7 sub-basins (B1 to B7), based on the river branches and
topographic futures. Fig. 1 shows the Barandoozchay map and hydrometeorological gauges.
This mountainous basin is mostly covered by grasslands, followed by farmland and orchard
land. The humid air often (not always) comes from the west, originating from the
Mediterranean Sea.
There are 6 daily rain gauges and 4 stream gauges inside the basin (Fig. 1), and 3 hourly
rain gauges (35010, 34013 and 34019) around the basin.

[Fig. 1 is here]

Seven typical storm events were selected during 1995 to 2014. These events have
recorded rain data (daily and hourly) available from the nearby rain gauges and the
hydrometric runoff data from the stream gauges.
**2.2 Estimation of sub-basin hyetograph**
When the cloud is stationary, most of sub-basins that are covered by the cloud react to the
rainfall simultaneously, implying that the start time and end time of the rainfall event is
approximately the same for all sub-basins; while in the case of a moving cloud, the sub-
basins that are located in the wind direction start to generate runoff earlier than the others
(Fig. 2).

[Figure 2 is here]

Since there is no record from the rain gauge inside the basin, the start and end time of the
events were unknown. Therefore, the residence time of the storm cloud over each sub-basin
and its role in outlet runoff generation were estimated and examined.




As the first step, the total daily rainfall of each sub-basin was estimated using Kriging
and IDW (Inverse Distance Weighted) methods, based on the rain gauges inside the basin.
Fig. 3 shows the raster map of generated rainfall for the event on May 12$^{th}$, 2010.
[Fig. 3 is here]
The total daily rainfall was then disaggregated into hourly rainfall. Since there is no
hourly recording gauge inside the basin, the nearest recording gauges at Urmia, Oshnavieh
and Naghadeh (35010, 34013 and 34019) were used. The hourly rainfall was obtained by
multiplying the estimated total daily rainfall by the ratio of hourly rainfall to the daily rainfall
(Choi, 2008; Gyasi-Agyei et al. 2005, 2007). Fig. 4 illustrates the procedures to disaggregate
the daily rainfall into each sub-basin's hyetograph.
[Fig. 4 is here]
Due to dynamic motion of the cloud, the rainfall duration, start and end time, and
intensity as well as other characteristics change. To determine the cloud arrival time of each
sub-basin, the recorded hyetograph was concentrated to a unique time named the Time of
Gravity Centre of Hyetograph (TGCH) (Khalighi 2009). Then the rainfall time over each sub-
basin TGCH was obtained through the following procedures:
(1) TGCH for recorded rainfall was calculated as a moment of the rainfall component
around the horizontal and vertical axis (Fig. 5). The figure shows that the recorded event in
station 35010 started at 4:00 am and ended at 2:30 pm, and the calculated TGCH was at 9:00
am (8.981).
(2) As there are only 3 recording gauges around the basin, a flat plane passes through the
stations (Fig. 6). Therefore, the equation of the plane (TGCH=aX+bY+c) was applied to
calculate the TGCH at each point (X,Y).
(3) The coordinates of the sub-basin centroids were placed in the above equations to
determine the TGCH of each sub-basin.





(4) The previously derived hyetograph was shifted as its gravity centre conformed to the
TGCH of each sub-basin centroid (Fig. 7).

[Fig. 5 is here]

[Fig. 6 is here]

[Fig. 7 is here]

For example, the TGCH for event 95/04/22 was recorded at 8.98, 6.48 and 5.33 at the
stations 35010, 34019 and 34013 respectively (table 2), then the equation of the TGCH plane
of this event was: $TGCH = 0.000077 * X + 0.000069 * Y - 317.457$. Based on this
equation and the coordinates of the B1 sub-basin centroid, the TGCH was 8:00 am, implying
that the TGCH at B1 occurred almost one hour earlier than at station 35010, which was 8:59
am.
**2.3 Rainfall-runoff modelling**
The HEC-HMS model (TR-55, 1986) was used to investigate the effects of storm movement
on hydrograph simulations. The model was calibrated by considering the most sensitive
parameters such as curve number ($CN$) and initial abstraction ($I_a$), via events 1995, 2002,
2003, 2006 and 2008. The validation was conducted using the events 2010 and 2014. After
the calibration and validation, the simulations were carried out for all events using two
hypotheses: (1) stationary cloud where the sub-basin hyetograph timing is equal to the nearest
recording gauge; (2) moving cloud where the sub-basin rainfall hyetograph shifted base on
cloud movement direction and sub-basin location.
A Taylor diagram (Taylor, 2001, 2005; Sigaroodi et al., 2014) and root mean squared of
relative difference (RD) were used to compare the results of two hypothesized conditions.

$$RD = \sqrt{\left(\frac{(P_O - P_S)}{P_O}\right)^2} * 100$$

where the $P_O$ and $P_S$ are observed and simulated peak discharge respectively.





## 3. Results


Fig. 8 shows the planes of TGCH for different events. Although the basin is mainly affected
by the humid Mediterranean air, the results indicated that each selected rainfall event had
unique characteristics.
[Fig. 8 is here]
Based on the gauge locations and TGCH of each event, a plane equation TGCH = aX +
bY + c was obtained for each event. Table 1 shows the equation coefficients.
[Table 1 is here]
The gravity centre coordinate of each sub-basin is used in the equations to calculate the
TGCH for the sub-basin centroid of each event. Fig. 9 shows how the sub-basin hyetograph is
shifted to obtain the TGCH for the event on April $3^{rd}$, 2003. The measured TGCH at the
gauges and the calculated TGCH for sub-basins are shown in Table 2.
[Fig.9 is here]
[Table 2 is here]
Fig. 10 presents the HEC-HMS modeled results for the event on April $22^{nd}$, 2014 at the
gauge 35005. The right part shows the model performance under stationary conditions where
all sub-basins react to the hyetograph simultaneously. The hydrograph is sharp and the time
to peak is quite different compared to the observed hydrograph. The left part presents the
modeled result using a shifted hyetograph, which matches better with the observed
hydrograph.
[Fig. 10 is here]
For comparison, the modeled peak discharges of the 7 selected events under the two
conditions are presented together with the observations in Table 3.
[Table 3 is here]





Fig.11 displays the standard deviation (SD) and correlation coefficient $R^2$ of the modeled
results under stationary and moving conditions on the Taylor diagram. It is clearly seen that
the moving condition results are closer to the observation points than the stationary condition
results.

[Fig. 11is here]

**4. Discussion**
To achieve accurate hydrological modeling, high quality and spatially-explicit rainfall data
should be accessible; however, in many cases uniform hyetographs are used for all sub-basins
due to lack of sufficient gauges. If the cloud motion is neglected, it means that the differences
between the times of runoff generation are ignored. In this case, to compensate for the
difference and achieve better matching between simulated and observed runoff, other basin
factors such as curve number (CN) have to be modified, which most probably cause artifacts
in the coefficients (Khalighi et al., 2006, 2009).
When the cloud movement is slow, consideration of movement is more important
compared to fast movement conditions. In the event of April 22nd, 2014, the time difference
between gauges 35010 and 34019 (Table 2) shows that the cloud movement is very low, thus
the sub-basin B1 generates runoff much earlier than B7. This result was not consistent with
the findings of Saghafian (1995), who stated that a stationary rainstorm could be used in low
speed storms. This study showed that for small basins or laboratory scales where the cloud
covers the whole basin, the storm motion effect can be ignored; while in the case of middle-
size to large basins, the runoff of low speed storms has an obvious role in determining
hydrograph shape. It can then be concluded that when the time difference between the
recorded rainfalls around the area is small, the differences between stationary and moving
runoff simulations are slight. These results were consistent with the findings of previous
studies, which showed the impacts of cloud motion on hydrographs by using rainfall



simulators at different laboratory scales (Sing, 1997, 1998; de Lima and Singh, 2002; de
Lima et al., 2003; Marzen, 2015) or the kinematic wave method (Mizumura, 2011).

The results of this study also revealed that longer rainfalls are less affected by cloud

movement. In other words, for rapid and short rains, the runoff hydrograph is more strongly
affected by cloud movement speed and direction. These results were consistent with the
findings of previous studies (de Lima and Singh, 2002; Khalighi, 2009; Dae-Hong Kim, 2013)
in laboratory.

However, it should be noted that the effects of cloud movement on hydrograph modeling

become visible only when the study area is divided into smaller sub-basins. In addition, a flat
plane is used to calculate the TGCH for the sub-basins in this study due to a lack of gauges;
but other interpolation methods such as IDW and Kriging could be more appropriate to obtain
surface data from the point data.

In conclusion, although there are many laboratory experiments on the effects of rainfall

movement on runoff simulation, more studies are necessary to determine how the spatial-
temporal dynamics of rainfall can be considered at the real watershed scale, in particular for
ungauged areas.





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





**Tables**

Table 1. Obtained coefficients for the TGCH flat plane

| Coefficient / Time | 95/04/22 | 02/04/21 | 03/04/03 | 06/04/18 | 08/04/07 | 10/05/12 | 14/04/22 |
|---|---|---|---|---|---|---|---|
| a | 0.000077 | 0.000256 | 0.000222 | 0.000244 | 0.000047 | -7.3E-05 | -8.9E-05 |
| b | 0.000069 | 0.000008 | 0.000095 | -3.4E-05 | -0.00003 | 0.000074 | -0.00019 |
| c | -317.457 | -144.736 | -485.298 | 30.743 | 127.119 | -236.65 | 855.542 |






Table 2. TGCH measured at the gauges and calculated for the sub-basins

| | | Precipitation Events | | | | | | |
|---|---|---|---|---|---|---|---|---|
| | Location | 95/04/22 | 02/04/21 | 03/04/03 | 06/04/18 | 08/04/07 | 10/05/12 | 14/04/22 |
| Gauges | 35010 | 8.98 | 20.4 | 7.3 | 14.7 | 3.0 | 8.3 | 15.9 |
| | 34019 | 6.48 | 26.7 [a] | 6.2 | 23.4 | 6.2 | 1.6 | 25.5 |
| | 34013 | 5.33 | 20.7 | 1.1 | 17.3 | 4.8 | 4.0 | 25.9 |
| Sub-basins | B1 | 8.0 | 21.1 | 6.3 | 16.1 | 3.7 | 6.8 | 18.7 |
| | B2 | 6.5 | 17.2 | 4.5 | 12.6 | 3.1 | 7.6 | 20.9 |
| | B3 | 5.6 | 16.6 | 3.7 | 12.5 | 3.3 | 6.9 | 23.2 |
| | B4 | 4.5 | 14 | 2.8 | 10.1 | 3 | 7.3 | 24.9 |
| | B5 | 4.4 | 14.8 | 2.9 | 11.1 | 3.3 | 6.7 | 25.5 |
| | B6 | 4.8 | 16.4 | 3.2 | 12.7 | 3.6 | 6.1 | 25.2 |
| | B7 | 6.6 | 20.5 | 5.1 | 16.3 | 4.1 | 5.6 | 22.3 |

a: The numbers over 24 refer to the next day.





Table 3. Modelled peak discharges under two conditions and differences

| Date | Hydrological Station | Peak Discharge | | | Difference (%) | |
| --- | --- | --- | --- | --- | --- | --- |
| | | Obs. | Stationary | Moving | Stationary | Moving |
| 2014/04/22 | 35005 | 297.9 | 352 | 315.3 | 18.2 | 5.8 |
| 2010/05/12 | | 34.8 | 31.5 | 34.4 | 9.5 | 1.1 |
| 2008/04/07 | | 61.4 | 70.15 | 65.6 | 14.3 | 6.8 |
| 2006/04/18 | | 96.15 | 100.5 | 100.13 | 4.5 | 4.1 |
| 2003/04/03 | | 20.1 | 20.4 | 20.3 | 1.5 | 1 |
| 2002/04/21 | | 65.9 | 42.9 | 41.6 | 34.9 | 36.9 |
| 1995/04/22 | | 37.45 | 51.2 | 42.58 | 36.7 | 13.7 |
| 2010/05/12 | 35003 | 12.2 | 14.4 | 13.4 | 18 | 9.8 |
| 2008/04/07 | | 51.9 | 65.16 | 63.4 | 25.5 | 22.2 |
| 2006/04/18 | | 85.4 | 93.8 | 93.57 | 9.8 | 9.6 |
| 2003/04/03 | | 3.7 | 3.5 | 3.8 | 5.4 | 2.7 |
| 2002/04/21 | | 24.3 | 28.8 | 26.1 | 18.5 | 7.4 |
| 1995/04/22 | | 113.2 | 127.7 | 127.3 | 12.8 | 12.5 |
| 1995/04/22 | 35001 | 83 | 83.3 | 83.3 | 0.4 | 0.4 |






# Figure captions



Figure 1. Barandoozchay basin and hydrometeorological gauges
Figure 2. Schematic of rainfall movement effect on runoff formation
Figure 3. Spatial distribution of rainfall event 2010/05/12
Figure 4. Schematic of rainfall hyetograph determination in sub-basin centroid. a) Daily
precipitation at nearest gauge, b) Hourly hyetograph at nearest gauge, c) Daily precipitation
in sub-basin centroid d) derived hyetograph for sub-basin
Figure 5. HYGC output for calculation of hyetograph centroid at 95/04/22 in station 35010
(Gx: Temporal coordinates of concentrated event, Gy: Average of incremental rainfall)
Figure 6. Flat plane passing through the TGCH for the event 1995/04/22
Figure 7. Shifting the hyetograph to the estimated TGCH
Figure 8. Precipitation time occurrence plane in different events
Figure 9. Hyetograph of sub-basins before shift (left) and after shift (right). (Red arrows show
the timing position of TGCH before and after shifting)
Figure 10. HEC-HMS output for rainfall event 2014/04/22, under two different conditions,
moving simulation (left) and stationary simulation (right)
Figure 11. Scatter plot of the simulated peak discharge for stationary and moving conditions
on a Taylor diagram









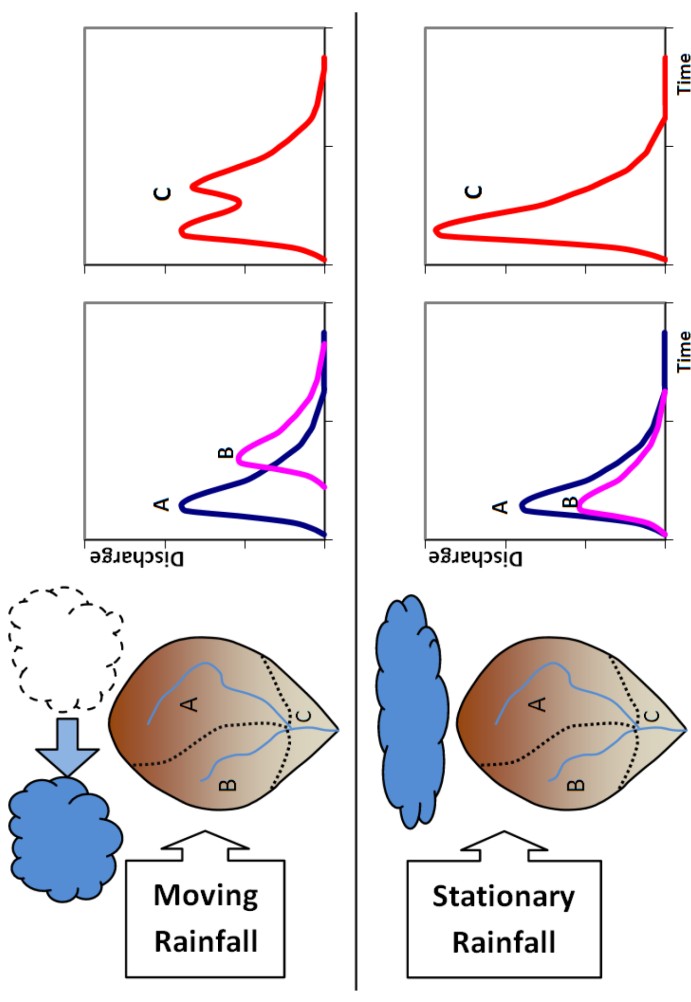



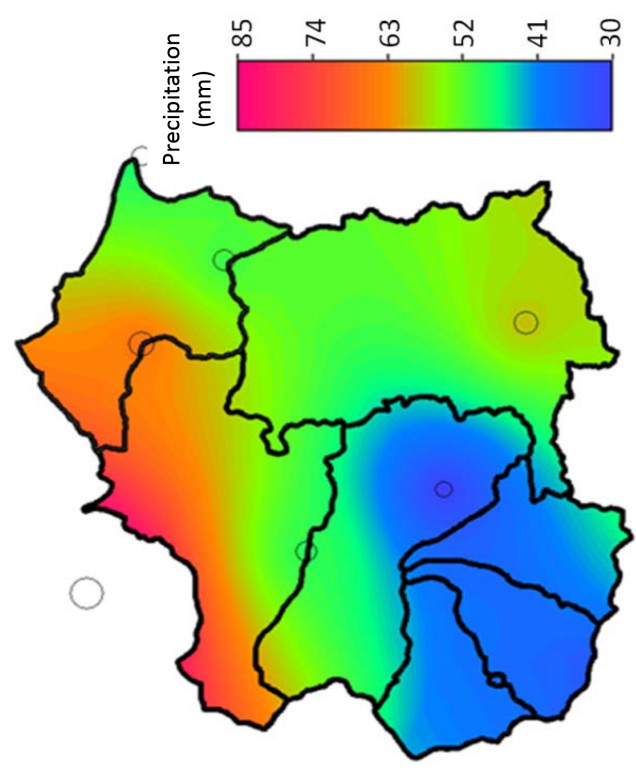





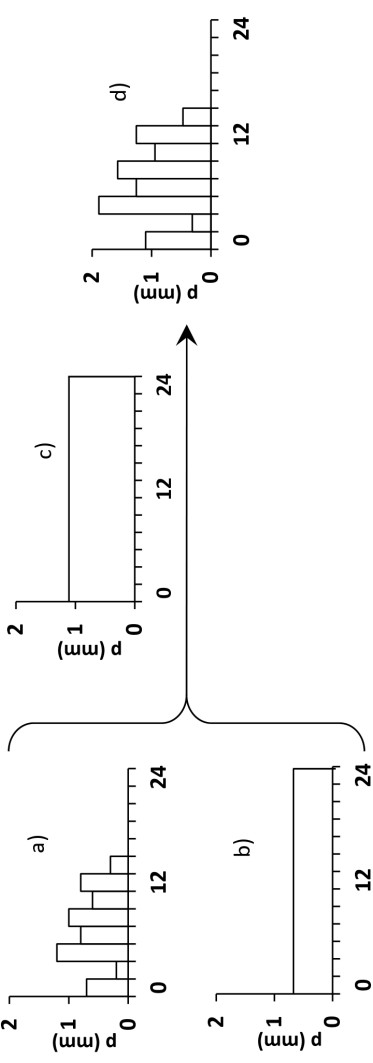





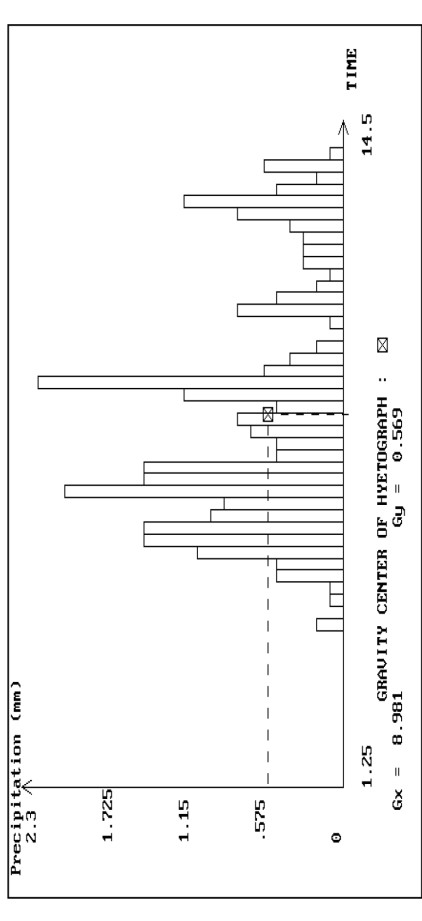





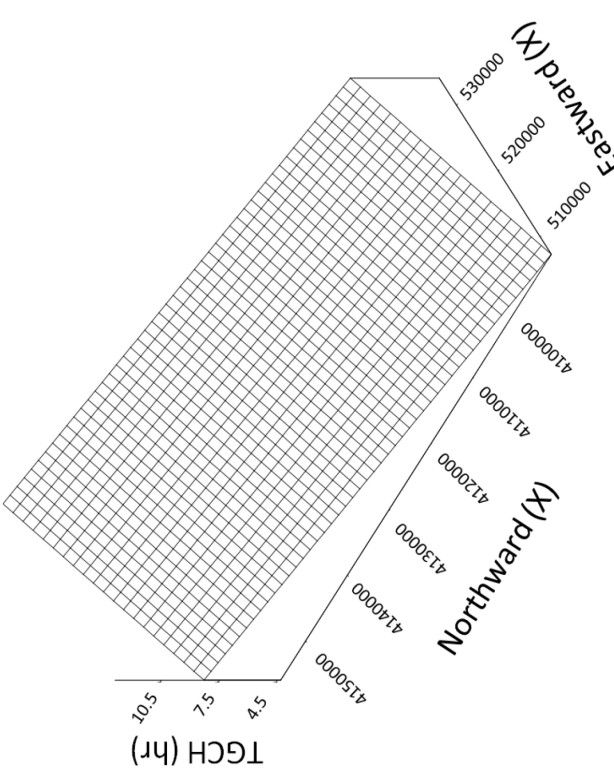



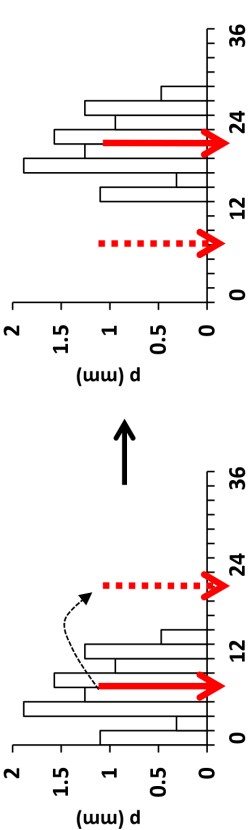



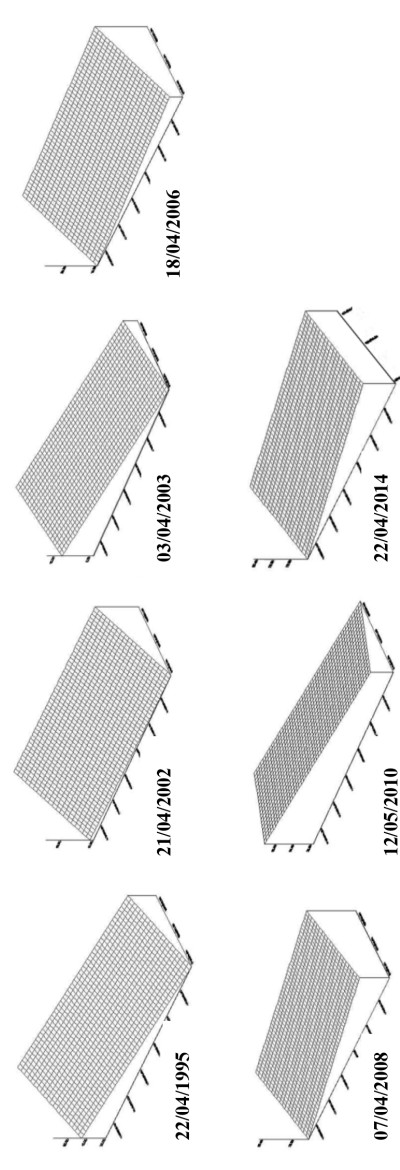




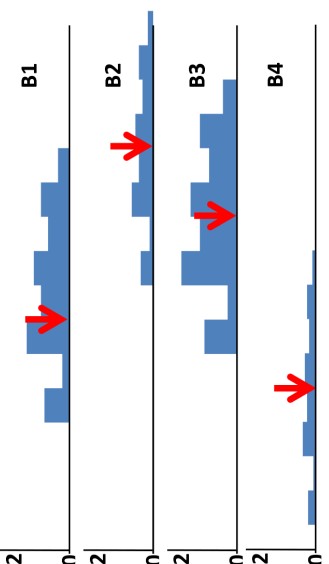

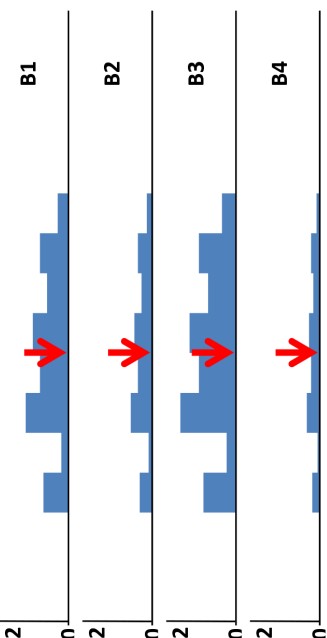





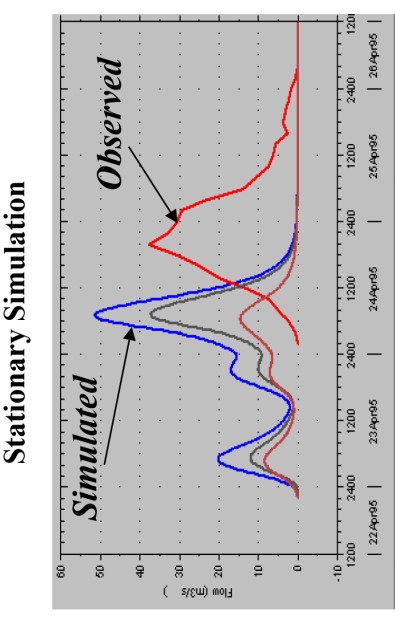

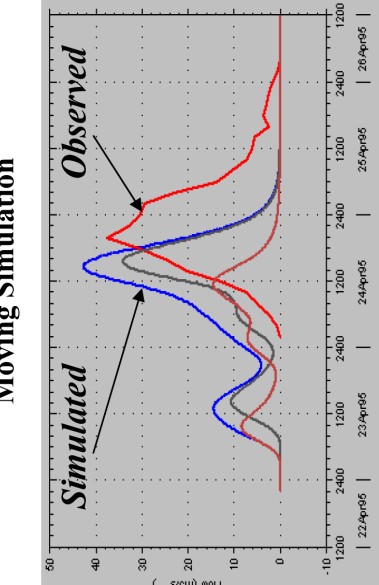

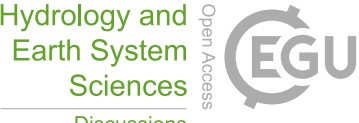



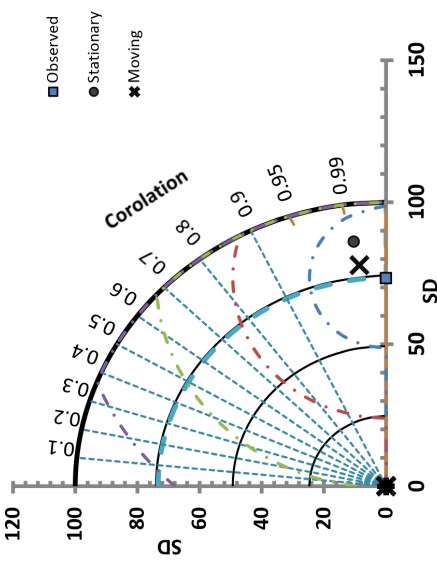