# Peer review of "Effects and consideration of storm movement in rainfall-runoff modelling at the basin"

_Hydrology and Earth System Sciences, 2016_

## Referee Comment (RC1) · G. Wang (Referee) · 22 Sep 2016

The manuscript presents a study on the effects of storm movement on rainfall-runoff modelling at the basin scale. The manuscript is generally well written with clear research goals and an appropriate structure to present the methods and results. Even though the goal of the research (precisely examine the effects of moving storms on hydrograph simulation) is extremely ambitious and the modeling results highly uncertain, I believe the authors are to be congratulated for focusing on an interesting topic and providing some critical answers on rainfall characteristics in ungauged basins. Some comments for this work are: 1. Line97-100, 'the hourly rainfall was obtained by multiplying the estimated total daily rainfall by the ratio of hourly rainfall to the daily rainfall'. How was this ratio calculated? Was it the mean ratio value of 3 gauges? How to make sure this ratio is appropriate for a certain daily rain gauge? 2. More information should

be presented on the concept of 'Time of Gravity Centre of Hyetograph'. How was it calculated and why it is important for determine the movement of cloud? 3. Is a linear equation appropriate for describing the TGCH plane? Is there any assumption made? (is the cloud pattern treated unchanged during the movement?) if so, the assumption should be explained in detail. 4. The calibration and validation of HEC-HMS model should be included. 5. The criteria for storm selection should be explained. What is 'typical storm events'? Why only 7 storms were selected? 6. More explicit conclusions of this research should be presented in the manuscript. 7. The figures are hard to understand, necessary explanations should be included.

---

## Referee Comment (RC2) · Anonymous Referee #2 · 4 Oct 2016

This manuscript evaluated the effects of rainfall movement on overland simulation through modelling the rainfall-runoff under stationary condition and moving condition at the basin scale. It is a very interesting topic, which can be beneficial to flood peak forecasing. And it is suitable for being published in Hydrology and Earth System Sciences. Also, it is well-organized. Therefore, I recommend being accepted after several minor revisions be made.

Detailed comments:

1. Legends for the figures are required. Also, more explanation for these figures should be added. And the differences among different color lines should be given more explanation in Fig.10.

2. Fig. 4 is interesting, however, more explanation are required, such as how to get

sub-figure d.

3. Line 142, what are the unique characteristics? Clarify please! In addition, the method for estimating the coefficients a, b, and c should be introduced.

4. The calibration and validation of HEC-HMS model should be described.

---

## Author Comment (AC1) · 20 Oct 2016

Manuscript ID: hess-2016-371 Full title: Study on the effects of storm movement on rainfall-runoff modelling at the basin scale Authors: Shahram Khalighi Sigaroodi, Qi-uwen Chen* ________________________________________________________________
Referee 1 The manuscript presents a study on the effects of storm movement on rainfall-runoff modelling at the basin scale. The manuscript is generally well written with clear research goals and an appropriate structure to present the methods and results. Even though the goal of the research (precisely examine the effects of moving storms on hydrograph simulation) is extremely ambitious and the modeling results highly uncertain, I believe the authors are to be congratulated for focusing on an interesting topic and providing some critical answers on rainfall characteristics in ungauged basins. –We thank the referee for this favorable assessment of our manuscript. We

are very grateful for his/her time and constructive comments on our manuscript. We have carefully considered all the comments and revised substantially the manuscript in accordance.

Some comments for this work are: 1. Line97-100, 'the hourly rainfall was obtained by multiplying the estimated total daily rainfall by the ratio of hourly rainfall to the daily rainfall'. How was this ratio calculated? Was it the mean ratio value of 3 gauges? How to make sure this ratio is appropriate for a certain daily rain gauge? –Thanks indeed for your important question, which helps us to present the work more clearly. The sentence was replaced by explanations in three steps. Please refer to line 100 -107.

2. More information should be presented on the concept of 'Time of Gravity Centre of Hyetograph'. How was it calculated and why it is important for determine the movement of cloud? – We are very grateful to this valuable comment, which improves the paper readability. More details and equation were added to the revised manuscript. Please refer to line 110 to 129.

3. Is a linear equation appropriate for describing the TGCH plane? Is there any assumption made? (is the cloud pattern treated unchanged during the movement?) if so, the assumption should be explained in detail. –Thanks indeed for this critical comment! Of course when cloud moves over a basin, the rainfall time at a point depends on the point location and cloud speed and direction. Although more stations could improve the accuracy, at least 3 gauges are necessary to record the rainfall to determine the occurrence time at a point. We added this explanation to the manuscript. Please refer to line 119 -122.

4. The calibration and validation of HEC-HMS model should be included. – Thanks for your great suggestions. More explanations and two tables were added to the revised manuscript about the calibration and validation of HEC-HMS model. Please refer to line 146-154 and new Tables 1 and 2.

5. The criteria for storm selection should be explained. What is 'typical storm events'?

Why only 7 storms were selected? –Thanks indeed for this essential point. Actually the phrase "typical storm event" refers to storm recording. For this study, we needed storms data that recorded in all three gauges around the area and also their runoff hydrograph records in hydrological stations. It means that among all storm events, those which were recorded in all stations were used. Hence to avoid misunderstanding the sentences was amended. Please refer to lines 78-80.

6. More explicit conclusions of this research should be presented in the manuscript. –Thanks indeed for this important and critical comment. It is essential to scientific publication. The most important value of the paper was explicitly added into the Discussion. Please refer to line 197 - 200.

7. The figures are hard to understand, necessary explanations should be included. –Thanks indeed for your valuable comment. More explanations were added to the manuscript for figures 4, 5 and 10. Please refer to lines 99-105, 117-121 and 180-181.

---

## Author Comment (AC2) · 21 Oct 2016

This manuscript evaluated the effects of rainfall movement on overland simulation through modelling the rainfall-runoff under stationary condition and moving condition at the basin scale. It is a very interesting topic, which can be beneficial to flood peak forecasting. And it is suitable for being published in Hydrology and Earth System Sciences. Also, it is well-organized. Therefore, I recommend being accepted after several minor revisions be made. –We thank the referee for this favorable and encouraging assessment on our manuscript. We are very grateful for his/her time and constructive comments on our manuscript. We have carefully considered all the comments and revised substantially the manuscript in accordance with these comments.

Detailed comments: 1. Legends for the figures are required. Also, more explanation for

these figures should be added. And the differences among different color lines should be given more explanation in Fig.10. –Thanks indeed for your careful comment. More explanations were added to the figures 4, 5 and 10. The figure 3 and 4 were redrawn. Please refer to lines 99-105, 117-121 and 180-181.

2. Fig. 4 is interesting, however, more explanation are required, such as how to get sub-figure d. –Thanks indeed for your great suggestion. This figure is very important to understand the whole paper. The Figure 4 was remade, and was explained step by step in revised manuscript. Please refer to line 98 -105.

3. Line 142, what are the unique characteristics? Clarify please! In addition, the method for estimating the coefficients a, b, and c should be introduced. –Thanks indeed for your important comment. The phrase was replaced by "different direction and speeds". Please refer to line 167. The method to calculate a, b and c was explained with reference to Howard (2010). Please refer to lines 124-126.

4. The calibration and validation of HEC-HMS model should be described. – Thanks for your great suggestions. More explanations and two tables were added to the revised manuscript about the calibration and validation of HEC-HMS model. Please refer to line 146-154 and new Tables 1 and 2.

---

## Author Comment (AC4) · 21 Oct 2016

The revised Fig. 3 and Fig. 4

[Figure]

[Figure]

**Fig. 1.** Figure 3. Spatial distribution of rainfall event 2010/05/12

[Figure]

**Fig. 2.** Figure 4. Schematic of rainfall hyetograph determination in sub-basin centroid. a) Hourly hyetograph at nearest gauge, b) Daily precipitation at nearest gauge, c) Daily precipitation in sub-basin cent

---

## Author Response (AR1)

**Manuscript ID**: HESS-2016-371

**Full title**: Effects and consideration of storm movement in rainfall-runoff modelling at the basin scale

**Authors**: Shahram Khalighi Sigaroodi, Qiuwen Chen
* * *
**Editor and referees comment:**

Enhance the discussion part! It should be more convincing and include the comparison of your results with previous studies and discuss the advantages of your method. The current comparison is not clear."

    We thank the editor and referees for their valuable comment. The following revisions have been made.

-- We did try to illustrate the advantages of the proposed method in "Introduction" and "Discussion" parts. In fact, there are few papers about the effects of storm movement on rainfall-runoff modeling under real conditions; therefore, the results are compared with the studies using laboratory or mathematical approaches, where the rainfall characteristics can be controlled. However, their results may not be applicable in real basins. The study provided a useful method to cope with rainfall movement in runoff modeling of sparsely gauged large watersheds.

-- To highlight the advantage. Extra contents are added. Please refer to line 19-21, 67-71, 204-212, 240-242.

-- The title was also changed to more precisely reflect the contribution of the paper.

-- Some mistakes in references were modified. Please refer to line 252, 256, 274 and 315.